# Country-wide medical records infer increased allergy risk of gastric acid inhibition

Galateja Jordakieva[1,2,3], Michael Kundi [4], Eva Untersmayr[2], Isabella Pali-Schöll[2,3], Berthold Reichardt[5] & Erika Jensen-Jarolim[2,3]

Gastric acid suppression promotes allergy in mechanistic animal experiments and observational human studies, but whether gastric acid inhibitors increase allergy incidence at a population level remains uncharacterized. Here we aim to assess the use of anti-allergic medication following prescription of gastric acid inhibitors. We analyze data from health insurance records covering 97% of Austrian population between 2009 and 2013 on prescriptions of gastric acid inhibitors, anti-allergic drugs, or other commonly prescribed (lipid-modifying and antihypertensive) drugs as controls. Here we show that rate ratios for anti-allergic following gastric acid-inhibiting drug prescriptions are 1.96 (95%CI:1.95–1.97) and 3.07 (95%-CI:2.89–3.27) in an overall and regional Austrian dataset. These findings are more prominent in women and occur for all assessed gastric acid-inhibiting substances. Rate ratios increase from 1.47 (95%CI:1.45–1.49) in subjects <20 years, to 5.20 (95%-CI:5.15–5.25) in >60 year olds. We report an epidemiologic relationship between gastric acid-suppression and development of allergic symptoms.

[1] Department of Physical Medicine, Rehabilitation and Occupational Medicine, Medical University of Vienna, Waehringer Guertel 18-20, 1090 Vienna, Austria. [2] Institute of Pathophysiology and Allergy Research, Center of Pathophysiology, Infectiology and Immunology, Medical University of Vienna, Waehringer Guertel 18-20, 1090 Vienna, Austria. [3] The Interuniversity Messerli Research Institute, University of Veterinary Medicine Vienna, Medical University Vienna, University of Vienna, Veterinaerplatz 1, 1210 Vienna, Austria. [4] Center for Public Health, Medical University of Vienna, Kinderspitalgasse 15, 1090 Vienna, Austria. [5] Sickness Fund Burgenland, Siegfried-Marcus-Straße 5, 7000 Eisenstadt, Austria. Correspondence and requests for materials should be addressed to E.J.-J. (email: erika.jensen-jarolim@meduniwien.ac.at)

A nti-ulcer drugs are intended for the prevention and treatment of acid-related disorders of the upper gastro-intestinal tract (GIT). PPIs reduce symptoms in gastro-esophageal reflux and dyspepsia or promote healing of peptic ulcers[1], and are applied for long-term usage such as in eosino-philic esophagitis, or as co-medication with NSAIDs such as in cardiovascular diseases or chronic pain. Their favorable safety profile has led to over-prescription by physicians[2], resulting in the fact that 44.9% of internal and 23.3% of surgical patients are already prescribed a PPI with hospital admission[3]. On the other hand, over-utilization by patients can be difficult to control, especially due to the dominating over-the-counter (OTC) avail-ability[4]. Little health risk was reported with short-term usage if applied according to label instructions, but the available data purely reflected prescription use[5]. In an ambulatory care study, only 35% of proton-pump inhibitor (PPI) prescriptions were based on an appropriately documented upper GIT diagnosis[6]. The view on PPI as harmless co-medication has increasingly been challenged by reports of potentially related complications, e.g., increased risk of osteoporotic fractures, *Clostridium difficile* or other enteric infections[7,8], pneumonia, and many more[9], espe-cially in long-term usage[10]. Over the past years, our group developed the concept that gastric acid inhibitors also promote the development of allergic disease not only in adults[11–16], but even imprinting the next generation for allergy[17]. Subsequent pregnancy, birth cohort studies, and meta-analyses fueled emer-ging concerns[18–20], reported also by pediatricians[21,22].

The common and desired effect of anti-ulcer medication is elevation of gastric pH by either blocking proton pumps or H2 receptors of gastric parietal cells, or direct binding gastric acid by aluminum compounds such as sucralfate. Alongside the mucosa-protective attributes of gastric pH elevation, pH-dependent pepsin activation for protein digestion is impaired, subsequently also affecting pancreatic digestion[13]. The persisting allergenic epitopes are large enough to trigger de novo sensitization via the intestinal mucosa and lead to specific IgE responses directed towards oral antigens, including nutritional proteins[14,15,23–25], drugs[14,26], and to the PPIs themselves[27–29]. Besides enabling the persistence of ingested epitopes and leading to antigen-specific Th2 type immune responses and allergic symptoms (Fig. 1a), a growing body of research indicates that anti-ulcer drugs may in an innate manner promote cellular responses towards a Th2 bias (Fig. 1b–e). For instance, PPIs activate mast cells via AhR[30] thereby synergizing with IgE-FcεRI signaling (Fig. 1b) and enhancing release of human mast cell mediators and CD63 expression associated with allergic symptoms[31]. H2 receptor antagonists (H2RA) stimulate Th2 cells to release Th2 cytokines that consequently promote the formation of IgE antibodies in humans, and additionally IgG1 in mice[31] (Fig. 1c). But they have also a Th2 promoting effect on monocytes, dendritic cells (DCs) and invariant natural killer cells (iNKTs)[32], especially in context with lipid antigen expressed with non-conventional antigen presentation molecules like CD1 (Fig. 1d). Several mouse studies underlined the Th2 promoting adjuvanticity effects of antacids in the absence of other adjuvants[11,23,24,33,34] (Fig. 1e). Notably, PPIs alter the gut and oral microbiome[35–37], which again plays an essential role in balancing the activity of Th2 cells[38], the key cellular players in IgE-mediated allergic disease (Fig. 1f). Our recent study in a mouse model corroborates the assumption that PPIs induce type 2 hypersensitivity via an impact on micro-biota[39]. The data suggest that by various antigen-specific, innate and adjuvant mechanisms anti-ulcer drugs shape a Th2 envir-onment making people prone to develop IgE-mediated hyper-sensitivity requiring anti-allergy medication.

Based on the current evidence from experimental and obser-vational human cohort studies, we here evaluate the prevalence of anti-ulcer medication in Austria and, using an epidemiological approach[40], assess any potential association with allergic symp-toms deduced from anti-allergic substance and desensitization prescriptions.

## Results

**Data collection in Austria.** An average population of 8.2 million people covered by the claims database of the health insurance companies provided 39,180,151 years of follow-up until pre-scription of an anti-allergic drug. Mean age was 42.2 years and 48.7% were males. During 8,133,846 years of follow-up after prescription of an acid inhibitor, 416,615 first prescriptions of an anti-allergic drug were registered (event rate: 5.12% person-years —%-PY). In contrast, 31,046,305 years of follow-up in those that had no acid inhibitor prescription during the observation period were associated with only 810,990 anti-allergic medications, corresponding to an event rate of 2.61%-PY. The rate ratio (ratio of the prescription rate in users of acid-inhibiting drugs to that in non-users) for prescription of an anti-allergic drug after acid-inhibiting medication in the total population was 1.96 (95% confidence interval—CI: 1.95–1.97) (Table 1). The prevalence for anti-allergic and acid-inhibiting drugs by gender and age groups is shown in Supplementary Table 1.

While both males and females had significantly increased prescription rates of anti-allergic substances after acid-inhibiting drug medications, males had significantly lower rate ratios (1.70 vs. 2.10 in females, $p < 0.001$ based on Wald chi-square test). There was also a highly significant age trend with rate ratios increasing from 1.47 (95% CI: 1.45–1.49) in those up to 20 years of age to 5.20 (95% CI: 5.15–5.25) for individuals above 60 years. However, incidence rate of anti-allergic medication was highest in young people and declined with increasing age (Table 1).

Even higher rate ratios were obtained when we contrasted individuals receiving an acid-inhibiting medication with those on antihypertensive or lipid-modifying (C09/C10) medication in the subsample from Burgenland (Table 2). Here, the overall rate ratio for an anti-allergic medication after treatment with an acid inhibitor was 3.07 (95% CI: 2.89–3.27). No significant difference was seen in this respect between genders, although, like in the data set for overall Austria, women had significantly higher prescription rates of anti-allergics (7.47%-PY vs. 5.18%-PY in males, $p < 0.001$ based on Wald chi-square test), especially those above the age of 20 (Fig. 2). Also in the analysis contrasting people with acid-inhibiting medications with those on C09/C10 treatment, an increase of rate ratios with increasing age was seen; however, the oldest age groups displayed no further increase. Similar to overall Austria, prescription rates of anti-allergic medications declined with increasing age (Table 2).

The prevalence of anti-allergic medications showed age and gender differences in accordance with previous reports (Fig. 2)[41]. Prescriptions steeply increased at young age, with an earlier peak in males (5–9 years) than in females (20–24 years). The prevalence declined in males from puberty to older age, while in females figures remained high at about 13% until a decline at 80 years and older. This trend is compatible with the declining incidence rates of anti-allergic medication prescriptions (Table 2) with increasing age. The prevalence of prescriptions of acid inhibitors, antihypertensive and lipid-modifying drugs are shown in the Supplementary Figs. 1–3.

Kaplan–Meier plots (Fig. 3) show that the cumulative fraction with an anti-allergic prescription seven years after starting an acid inhibitor medication is ~37%, while in those with a C09/C10 prescription this fraction amounts to background levels (~14%). The hazard ratio obtained in Cox regression adjusted for age and gender was 2.05 (95% CI: 1.91–2.19), while the unadjusted hazard

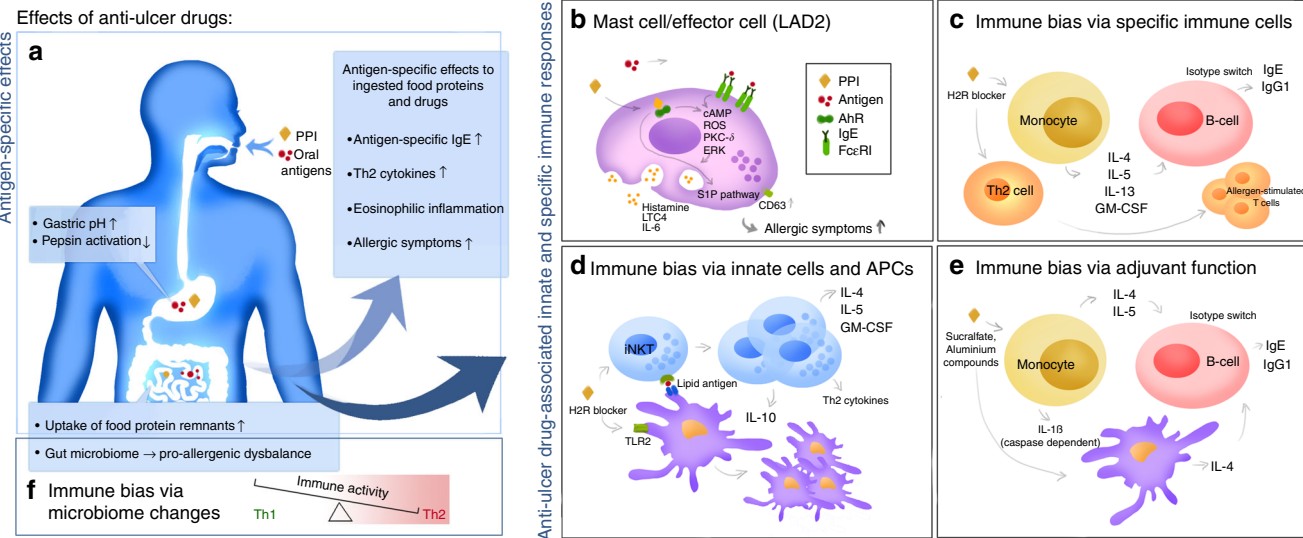

**Fig. 1** Overview of direct and indirect pro-allergenic immune responses to anti-ulcer drugs (AUD). **a**, **f** With regards to oral allergens, the gastric pH elevation by AUDs, most dominantly by proton-pump inhibitors (PPI) and H2 receptor antagonists (H2RA), leads to reduced pepsin activation and impaired food antigen degradation, enabling persistence of ingested epitopes and their uptake in the intestines[14–16,23,24]. This leads to formation of antigen-specifc IgE and promotion of a Th2 type dominated immune milieu resulting in eosinophilic inflammation and allergic symptoms[14,15,24]. **b–e** With regards to directly AUD-associated innate and adjuvant immune effects, PPIs can (**b**) induce AhR-mediated mast cell activation synergizing with IgE-FcεRI signaling and resulting in mediator release[30] and enhanced CD63 expression via the S1P pathway; both mechanisms result in allergic symptoms manifestation; (**c**) H2RAs stimulate the release of Th2 cytokines from both monocyte and Th2 cells leading to a B-cell isotype switch resulting in IgG1 (mice) and IgE production (humans, mice)[31]; (**d**) Further, H2RAs promote an increase in Th2 cytokine releasing iNKT cells[32], particularly when co-exposed to lipid antigens, and increase in CD1d+ DCs via TLR2 activation; (**e**) Sucralfate, an aluminum compound, can induce IL-4 release from DCs along with IL-5[33,34] and caspase-dependent IL-1β secretion[43] from monocytes promoting an isotype switch in B-cells into IgE production; **f** The resulting disturbance of gut microbiota composition[35,37,45] additionally supports a pro-allergic Th2 milieu and enhances allergic symptoms

**Table 1 Incidence of anti-allergic drug prescription in patients with vs. without previous gastric acid inhibitor prescription**

|  | Person-years | # with anti-allergic prescription | Incidence rate % | Rate ratio | *p*-value |
|---|---|---|---|---|---|
| Persons with previous gastric acid inhibitor prescription | | | | | |
| Total | 8,133,846 | 416,615 | 5.12 (5.11–5.14) | 1.96 (1.95–1.97) | <0.001 |
| Males | 3,455,147 | 143,942 | 4.17 (4.14–4.19) | 1.70 (1.69–1.71) | <0.001 |
| Females | 4,678,699 | 272,673 | 5.83 (5.81–5.85) | 2.10 (2.09–2.11) | <0.001 |
| Age < 20 y | 311,203 | 20,971 | 6.74 (6.65–6.83) | 1.47 (1.45–1.49) | <0.001 |
| 20–39 y | 1,599,819 | 94,640 | 5.92 (5.88–5.95) | 2.05 (2.03–2.06) | <0.001 |
| 40–59 y | 3,086,748 | 151,897 | 4.92 (4.90–4.95) | 2.79 (2.77–2.81) | <0.001 |
| ≥60 y | 3,136,076 | 149,107 | 4.76 (4.73–4.78) | 5.20 (5.15–5.25) | <0.001 |
| Persons without previous gastric acid inhibitor prescription | | | | | |
| Total | 31,046,305 | 810,990 | 2.61 (2.61–2.62) | | |
| Males | 15,742,139 | 386,094 | 2.45 (2.44–2.46) | | |
| Females | 15,304,166 | 424,896 | 2.78 (2.77–2.78) | | |
| Age < 20 y | 7,792,981 | 357,776 | 4.59 (4.58–4.61) | | |
| 20–39 y | 8,469,564 | 244,656 | 2.89 (2.88–2.90) | | |
| 40–59 y | 8,624,276 | 152,044 | 1.76 (1.75–1.77) | | |
| ≥60 y | 6,159,484 | 56,514 | 0.92 (0.91–0.92) | | |

Incidence rates and 95% confidence intervals (CI) of anti-allergic drug prescriptions in individuals with and without acid inhibitor medication from data of overall Austria and rate ratios for an anti-allergic drug prescription in individuals with acid inhibitor medication relative to those without. *p* values based on Wald chi-square test

ratio (2.95, 95% CI: 2.77–3.14) was close to the rate ratio (Table 3). Restricting the analysis to those individuals that had a hospitalization due to dyspepsia, gastritis/duodenitis or gastric/duodenal ulcer (overall 2022 individuals) led to almost the same hazard ratios: adjusted for sex and age: 2.09 (95% CI: 1.37–3.22, *p* < 0.001 based on Cox chi-square test), unadjusted: 2.26 (95% CI: 1.41–3.62, *p* < 0.001 based on Cox chi-square test).

Individuals receiving an acid inhibitor had a higher proportion of C09/C10 prescriptions (44.3%) compared to those without (20.1%). However, this difference was almost entirely due to co-medication with a non-steroidal anti-inflammatory drug (M01), and the percentage of C09/C10 prescriptions is similar to the rate in those without an acid inhibitor prescription if restricted to those without M01 co-medication (20.7%) (Supplementary

**Table 2 Incidence of anti-allergic drug prescription in patients after prescription of gastric acid inhibitors or control medications**

| | Person-years | # with anti-allergic prescription | Incidence rate % | Rate ratio | p-value |
|---|---|---|---|---|---|
| Persons with previous gastric acid inhibitor prescription | | | | | |
| Total | 117,287 | 7706 | 6.57 (6.42–6.72) | 3.07 (2.89–3.27) | <0.001 |
| Males | 46,002 | 2383 | 5.18 (4.97–5.39) | 2.93 (2.66–3.22) | <0.001 |
| Females | 71,285 | 5323 | 7.47 (7.27–7.67) | 2.92 (2.69–3.17) | <0.001 |
| Age < 20 y | 9,543 | 1058 | 11.09 (10.42–11.75) | 2.04 (1.20–3.45) | 0.008 |
| 20–39 y | 48,502 | 3636 | 7.50 (7.25–7.74) | 2.41 (2.01–2.91) | <0.001 |
| 40–59 y | 47,097 | 2467 | 5.24 (5.03–5.44) | 2.52 (2.30–2.77) | <0.001 |
| ≥60 y | 12,145 | 545 | 4.49 (4.11–4.86) | 2.22 (1.96–2.51) | <0.001 |
| Persons with previous C09/C10 prescription | | | | | |
| Total | 53,793 | 1151 | 2.14 (2.02–2.26) | | |
| Males | 28,549 | 505 | 1.77 (1.61–1.92) | | |
| Females | 25,244 | 646 | 2.56 (2.36–2.76) | | |
| Age < 20 y | 257 | 14 | 5.45 (2.59–8.30) | | |
| 20–39 y | 3,704 | 115 | 3.10 (2.54–3.67) | | |
| 40–59 y | 26,190 | 544 | 2.08 (1.90–2.25) | | |
| ≥60 y | 23,642 | 478 | 2.02 (1.84–2.20) | | |

Incidence rates and 95% confidence intervals (CI) of anti-allergic drug prescriptions in individuals with acid inhibitor medication and with anti-hypertensive or lipid-modifying (C09/C10) drug prescriptions from data of the Austrian county Burgenland and rate ratios for an anti-allergic prescription in individuals with acid inhibitor medication relative to those with C09/C10 prescriptions. p values based on Wald chi-square test

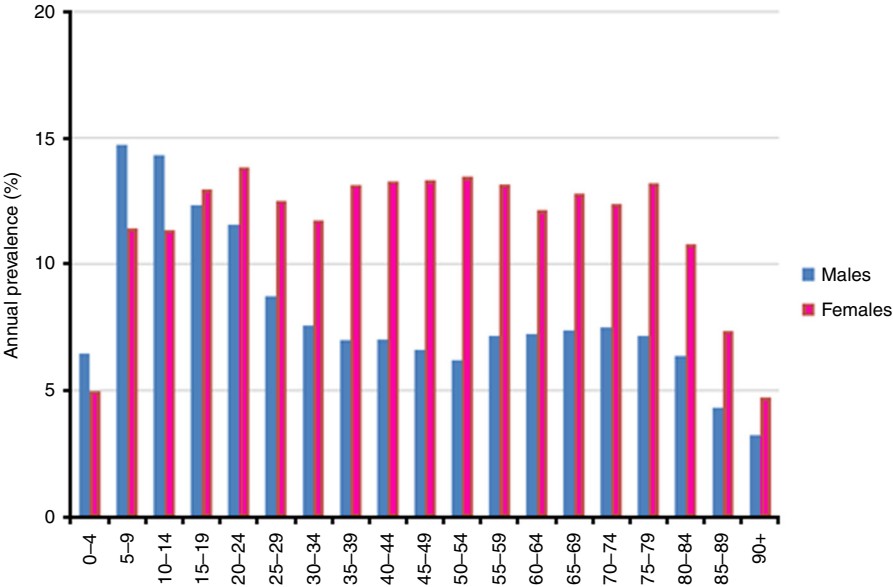

**Fig. 2** Annual prevalence of anti-allergic drug prescriptions. Data presented by gender and age groups 2009–2013 in the Austrian county Burgenland with 977,488 person-years of follow-up

Table 3). Risk of a subsequent prescription of an anti-allergic drug was, however, unaffected comparing those with and without M01 co-medication (Supplementary Fig. 4).

In essence, the higher hazard for prescription of an anti-allergic medication is present for all groups of acid inhibitors (Table 3, Supplementary Fig. 5), with the exception of individuals on prostaglandin E2, whose numbers were too low to draw clear conclusions. These results did not differ significantly whether the first or the longest prescription was assessed. Restricting analyses to those that had only one type of acid-inhibiting prescription led to the following hazard ratios (adjusted for sex and age): PPI: 2.45 (95% CI: 2.15–2.79), H2RA: 2.40 (95% CI: 2.11–2.72), Su: 1.97 (95% CI: 1.85–2.10) (Supplementary Fig. 6). There was a clear hazard increase (p for trend <0.001) with increasing number of daily doses per year, but already the lowest quartile (up to 20 daily doses per year) had a significantly increased hazard ratio

(adjusted HR: 1.28, 95% CI: 1.18–1.39). Interestingly, there was no further hazard increase above the third quartile (Table 3, Supplementary Fig. 7). Trend analysis allowed estimation of a cut-off for the number of annual doses leading to a significantly earlier prescription of an anti-allergic medication. This cut-off was determined at 6 daily doses (Supplementary Fig. 8).

## Discussion

Revealing consistent relationships between exogenous factors and medical conditions may have a broad impact on public health[40]. In our population-based analysis of claims data, covering nearly all of Austria's population (8.2 million) between 2009 and 2013, we found a high prevalence of anti-ulcer drug prescription associated with a highly significant risk elevation for subsequent prescription of anti-allergic medications. This association was even clearer when comparing the prescription frequency of

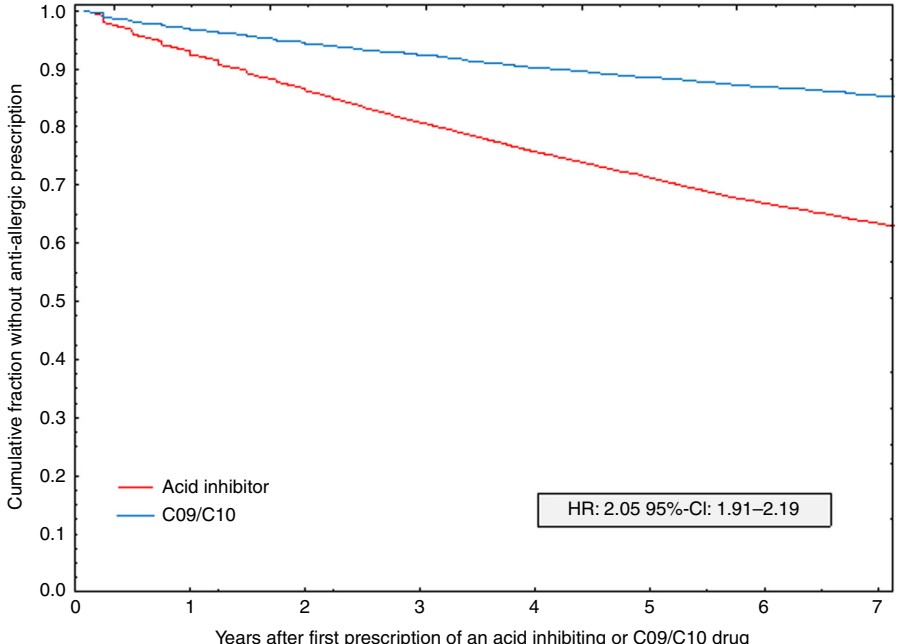

**Fig. 3** Anti-allergic medication following treatment with acid-inhibiting drugs and controls. Kaplan–Meier plots of the cumulative proportion without subsequent first anti-allergic medication for those with an acid-inhibiting drug prescription and the contrast group of those with an anti-hypertensive or lipid-modifying drug (C09/C10) prescription

| Table 3 Hazard ratios for prescription of anti-allergic drugs after prescription of gastric acid inhibitors or control medications | | | | | | | |
|---|---|---|---|---|---|---|---|
| Group | N | HR | 95% CI | *p*-value | aHR | 95% CI | *p*-value |
| Total | 36,608 | 2.95 | 2.77–3.14 | <0.001 | 2.05 | 1.91–2.19 | <0.001 |
| Males | 16,070 | 2.78 | 2.52–3.06 | <0.001 | 1.95 | 1.75–2.17 | <0.001 |
| Females | 20,538 | 2.81 | 2.59–3.05 | <0.001 | 2.09 | 1.91–2.29 | <0.001 |
| Age < 20 y | 2972 | 2.02 | 1.19–3.42 | 0.009 | 1.93 | 1.14–3.28 | 0.014 |
| 20–39 y | 13,054 | 2.41 | 2.00–2.90 | <0.001 | 2.20 | 1.82–2.65 | <0.001 |
| 40–59 y | 14,812 | 2.45 | 2.23–2.68 | <0.001 | 2.25 | 2.04–2.47 | <0.001 |
| ≥60 y | 5770 | 2.14 | 1.89–2.42 | <0.001 | 2.07 | 1.83–2.34 | <0.001 |
| First prescribed acid inhibitor | | | | | | | |
| H2RA | 2675 | 2.95 | 2.70–3.22 | <0.001 | 2.04 | 1.86–2.23 | <0.001 |
| PGE2 | 8 | 3.41 | 1.10–10.59 | 0.034 | 2.40 | 0.77–7.47 | 0.129 |
| PPI | 24,309 | 2.93 | 2.75–3.12 | <0.001 | 2.05 | 1.91–2.19 | <0.001 |
| SU | 641 | 3.72 | 3.21–4.32 | <0.001 | 2.14 | 1.84–2.50 | <0.001 |
| Longest applied acid inhibitor | | | | | | | |
| H2RA | 1554 | 3.43 | 3.08–3.82 | <0.001 | 2.34 | 2.10–2.62 | <0.001 |
| PGE2 | 7 | 3.52 | 0.88–14.10 | 0.075 | 2.66 | 0.66–10.64 | 0.167 |
| PPI | 25,465 | 2.90 | 2.72–3.08 | <0.001 | 2.03 | 1.90–2.17 | <0.001 |
| Su | 607 | 4.02 | 3.45–4.69 | <0.001 | 2.28 | 1.95–2.68 | <0.001 |
| Daily acid inhibitor dose per year | | | | | | | |
| <21 DDD/y | 6906 | 1.95 | 1.80–2.10 | <0.001 | 1.28 | 1.18–1.39 | <0.001 |
| 21–68 DDD/y | 6910 | 2.52 | 2.33–2.72 | <0.001 | 1.76 | 1.63–1.91 | <0.001 |
| 68–213 DDD/y | 6907 | 3.92 | 3.65–4.20 | <0.001 | 2.67 | 2.47–2.88 | <0.001 |
| ≥213 DDD/y | 6909 | 3.69 | 3.44–3.97 | <0.001 | 2.57 | 2.38–2.78 | <0.001 |

Hazard ratios (HR) and 95% confidence intervals (CI) from Cox regression and hazard ratios adjusted for gender or age or both (aHR) by subgroups of subjects with acid inhibitor prescriptions against individuals with anti-hypertensive or lipid-modifying medications (C09/C10 codes of ATC/DDD). *p* values based on Cox chi-square test
*H2RA* H2-receptor antagonists, *PGE2* prostaglandine E2, *PPI* proton-pump inhibitors, *SU* sucralfate, *DDD* defined daily dose, equal to the number of days the prescribed units would last, if completely consumed

anti-allergic medication in patients previously treated with acid-inhibiting drugs to those who had received other commonly used medications (antihypertensive or lipid-modifying substances), with an overall rate ratio of 3.07 [95% CI: 2.89–3.27]. While acid inhibitor treatment increased prescription rates of anti-allergic medication regardless of gender, women were more likely than men to receive anti-allergic medication in general, as well as after

the use of acid inhibitors. This may reflect the female dominance in IgE-mediated allergies[41]. Despite the generally higher prescription rates of anti-allergic medication in younger patients[2], the association between acid-inhibiting and anti-allergic medication was even more prominent with increasing age, consistent with a steady risk elevation with rate ratios ranging from 1.47 (95% CI: 1.45–1.49) in the young (<20 years) to 5.20 (95% CI:

5.15–5.25) in elderly patients (>60 years). The use of anti-acids in a geriatric cohort has previously been shown to be associated with elevated IgE and ST2 levels, a marker for Th2 responses[25].

Since all analyzed acid inhibitor drug classes (PPI, sucralfate, H2-receptor antagonists, prostaglandin E2) correlated with increased prescription rates for anti-allergic medication, the mechanism appears to be based on gastric pH modulation in general rather than on a particular drug-specific mode of action. A significantly higher prevalence of anti-allergic drug use after prescription of sucralfate, a sucrose sulfate aluminum salt, was noted. Aluminum compounds including sucralfate act as Th2 adjuvants[33,34]. Since patients are often prescribed different types of acid-inhibiting drugs in succession, restricting analyses to those that received only one type of drug allows a better comparison with the acid-inhibiting potency. PPIs and H2RA are thought to have similar potency, while sucralfate has a lower potency[42] in accordance with the hazard ratios for subsequent prescription of an anti-allergic medication.

It has to be noted that, although rare, these findings likely also include antihistamine prescriptions for allergic responses elicited by the anti-ulcer or hypertensive/lipid-modifying substances themselves; allergic sensitization to orally administered drugs resulting from preceding pH modulation by anti-ulcer drugs has been previously described[14,26].

Further, the cumulative dose of anti-ulcer drug use appears to have an impact on the subsequent need for anti-allergic medication. Although the daily acid inhibitor drug doses per year were associated with a clear increase in hazard ratios plateauing above the third quartile, as little as six daily doses per year were sufficient to increase the risk for subsequent anti-allergic medication use. This finding implies that the underlying effect is unleashed early on after first acid-inhibiting drug utilization.

Since acid inhibitors are a very commonly prescribed drug class not only in Austria but in developed countries worldwide, we aimed to rule out the bias of multi-medication. Therefore, we assessed the risk for anti-allergic substance prescriptions after consumption of other allergy-unrelated common drug classes, namely lipid-modifiers (e.g., statins) and antihypertensive substances. In a regional subgroup analysis, this more comprehensive assessment was possible since data also for these drugs were available. We found no risk elevation for anti-allergic medication after use of the C09/C10 drug subclasses. These findings imply an increase of allergic symptoms with a resulting need for symptom-relieving drugs, specifically after the use of acid-inhibiting drugs, but not other common drug classes.

Previous mechanistic studies in murine models have shown the development of de novo type I allergic sensitizations to dietary compounds[11,17,23,24], and to oral drugs[14] following acid inhibitor treatment. An observational study in 152 human adults further confirmed an impact of anti-ulcer drugs on boosting pre-existing, and inducing novel IgE specificities[15], and underlined their clinical relevance[23]. A mechanistic explanation for the allergy promoting effect of acid inhibitors is the hampered protein degradation resulting from the therapeutic elevation of gastric pH. This results in the persistence of otherwise labile and harmless nutritional proteins during gastrointestinal transit, which elevates their allergenic potential[13].

Research data emerging over the last decades further indicate that anti-ulcer drugs may directly promote allergic symptoms. The reported underlying mechanisms include immune bias of adaptive cellular responses to oral antigens (Fig. 1a), but also innate and adjuvant effects of the drugs themselves (Fig. 1b–e), with implications for a systemic Th2 bias. In mice, H2RAs support CD1d-mediated presentation of lipid antigens to iNKT cells, elevating numbers of both CD1d + DCs and iNKT cells; they were also shown to induce IL-4, IL-5, and GM-CSF release from monocytes and to aid Th2 cytokine secretion[32]. The H2RA cimetidine has been shown to enhance Th2 responses, potentially contributing to IgE production and thus allergic symptom manifestation[31]. Sucralfate has also been shown to act as a Th2 adjuvant even though it is administered orally[33,34]. Aluminum, a compound of sucralfate, further induces IL-4 release enabling isotype switch in B-cells leading to additional IgE secretion and caspase-dependent IL-1β secretion[43]. In humans, previous data indicate that omeprazole (a commonly prescribed PPI) can lower the threshold for mediator release in mast cells via transcription factor aryl hydrocarbon receptor (AhR) activation[44], resulting in lowering the allergen thresholds needed for eliciting allergic symptoms in sensitized patients. Additional changes in gut microbiome composition induced by anti-ulcer drugs and the associated dysbiosis and skewing of Th2 cell activity into a pro-allergenic state[38], further contribute to the establishment of a Th2 dominated immune response and enhanced manifestation of allergic symptoms, altogether further promoting an increased demand for pharmacological symptom relief.

In conclusion, in this population-based study covering claims data from all Austria and more comprehensive regional data of one county, which is the largest of its kind, we observed a highly significant increase in prescription of drugs relieving allergic symptoms in patients who were on treatment with gastric acid inhibitors of any class. Our findings confirm an epidemiological association between gastric acid suppression and development of allergic symptoms, in line with previous mechanistic animal trials and human observational studies. The evidence provided herein concurs with the emerging concern in terms of adults[10,26] and especially of pregnant females treated with antacid drugs[18,20-22].

There are some limitations to our study. First, we included antihistamines and substances used in desensitization therapy as anti-allergic medication, but not other related drug classes, e.g., corticosteroids. Despite their application in severe allergic reactions, corticosteroids have a broad range of other therapeutic indications beyond the treatment of allergic symptoms, unlike antihistamines, which are nearly exclusively used in the context of allergic symptom relief. Secondly, our data reflect prescription and purchase frequency of the respective drug classes, but actual intake frequency and duration could not be monitored within this study. However, patient compliance is expected to largely equally affect medication with all of the analyzed substance classes, as side effects for all substance classes are equally limited. Thirdly, the commonly prescribed anti-hypertensive medications (e.g., angiotensin-converting enzyme [ACE] inhibitors, angiotensin-receptor blockers) used in the control medication in our study, have specific side-effects and absolute contraindications (pregnancy, breastfeeding). A further source of bias could be that individuals with acid-inhibitor prescriptions are generally more likely to receive any medication. As we had no access to all types of prescriptions this could only partially be tested: we found a higher proportion of prescriptions of lipid-modifiers and anti-hypertensive substances (C09/C10) in those on acid-inhibitors as compared to those without. However, excluding co-prescription of acid-inhibitors and non-steroidal anti-inflammatory drugs, there was virtually no difference in subsequent C09/C10 prescriptions between acid-inhibitor users and non-users (20.7% and 20.1%, respectively). Due to restrictions of data protection we had no access to individual data such as co-morbidities, diagnoses and prescriptions other than those directly related to the research question. It was, therefore, not possible to perform a propensity score matching or adjustment and only gender and age were used as covariates. Since we have no data about indication of acid suppression, there is a possibility that overlapping symptoms (such as cough) may have led to both, an anti-acid and an

anti-allergic, medication. While this may be rarely the case, our analysis of the risk for subsequent anti-allergic prescriptions restricted to those hospitalized for dyspepsia, gastritis/duodenitis, or gastric/duodenal ulcer revealed virtually the same hazard as obtained for the total population. Therefore, lack of data on indication is unlikely to have biased our findings.

## Methods

**Data set from overall Austria.** Claims data from all Austrian compulsory health insurance companies were compiled. The Austrian health insurance system provides a nearly complete health care coverage for all residents according to their respective region of residence and current or former employment. Data handling and storage are in strict agreement with laws of privacy; data is pseudonymized for privacy preservation and retrieved using SAS 9.3 software (SAS Institute Inc., North Carolina). The health insurance databases store outpatient and inpatient medical service data covered by the health insurance funds, including demographics, hospital discharges along with ICD-system (International Classification of Diseases) coded primary diagnoses; the data base also contains all refundable prescriptions for all insured individuals (about 97% of the Austrian population, 8.1 to 8.3 million people during 2009–2013). Selection criteria were any prescription for a gastric acid-inhibiting drug (PPI; sucralfate, SU; H2-receptor antagonists, H2RA; or prostaglandin E2, PGE2, subsequently summarized as acid inhibitors) and/or any prescription of anti-allergic medications (antihistamines or specific allergen immunotherapy [AIT]) during the years 2009 to 2013. The endpoint of data collection at the end of 2013 was chosen due to the drop of PPI costs below the prescription fee in Austria thereafter, resulting in an incomplete registry of gastric acid-inhibiting drug data. Data extracted were: birth date; gender; insurance provider; medication by substance classes and subclasses of the WHO Anatomical Therapeutic Chemical Classification System with Defined Daily Doses (ATC/DDD) (Supplementary Table S2); dates of first and last prescription; total packages, and daily doses for every year of follow-up. In addition, numbers of insured individuals during the whole period by 5-year age groups and gender were provided for each year of follow-up. No imputation methods for missing data were applied. Missing data were extremely rare and were about 1 in 100,000. There was no missing data regarding age as date of birth is part of each individual insurance number. Daily dosage and prescription dates are also obligatory data for any drug prescription. Single individual "over the counter" antacid products are not included in the data set, however these products have very low pH modulating potency, are sold in small package sizes and are high-priced rendering their potential impact negligible. The study was approved by the ethics committee of the Medical University Vienna, Austria, prior to commencement (trial registration ECS 1134/2014).

**Data set from the regional sickness fund of Burgenland.** From the Austrian county Burgenland, the same claims data as from overall Austria but in addition all prescriptions for antihypertensive (i.e., "agents acting on the renin–angiotensin system") and/or lipid-modifying agents (codes C09 and C10 of the ATC/DDD system) during the observation period were provided. Data from this insurance company "Burgenländische Gebietskrankenkasse, BGKK" cover between 189,500 individuals in 2009 and 202,420 in 2013, providing overall 977,488 person-years of follow-up. Both data sets were provided by the insurance companies. No data were excluded from the data bases.

**Statistical evaluation.** Prevalence of drug prescriptions were computed by gender and 5-years age group (and year of follow-up) by counting subjects that had any prescription during the respective period in relation to the number of insured subjects of the same group during this period as the denominator.

From data of overall Austria for all individuals with a prescription of an acid inhibitor that had previously not been prescribed an anti-allergic medication, the total person-years of follow-up until prescription of an anti-allergic drug or end of follow-up (12-31-2013), whichever came first, was determined and the incidence rate (per percent person-years) of an anti-allergic prescription was computed. From the total person-years of follow-up the person-years of those with an acid inhibitor prescription was subtracted (including the period of an anti-allergic prescription and the period between the beginning of follow-up and first prescription of an acid inhibitor) and formed the denominator for computation of the incidence rate of an anti-allergic prescription for those without an acid inhibitor prescription. Confidence intervals were computed based on the Poisson distribution. These calculations were performed overall and stratified by gender and age groups. Comparisons of rate ratios were done by Wald chi-square tests.

We considered a possible bias that people under treatment for any disease or medical condition could be more likely to receive earlier treatment for different conditions including allergy. Therefore, a more comprehensive analysis was conducted based on insured persons from a region of Austria for which also data of a control group were available, namely individuals that had any prescription of an antihypertensive and/or lipid-modifying agent (codes C09 and C10 of the ATC/DDD). In addition to an analogous computation as described for overall Austria, we performed Cox regression analyses of those with a C09/C10 prescription during 2009 to 2013, but no acid inhibitor prescription since 2005 as reference. Time to occurrence of an anti-allergic medication or end-of-follow-up was the dependent variable in this case. Analyses were performed overall, and stratified by gender and age groups. Furthermore, acid inhibitor prescriptions were grouped by type of first prescription and type of longest prescription and an additional analysis was performed for daily doses per year of overall acid inhibitors grouped by quartiles. These analyses were adjusted for gender and age.

Since it is possible that not acid-inhibitors as such but an underlying disease could be related to the risk of an anti-allergic prescription, we performed Cox regression of time to anti-allergic prescription with previous anti-acid prescription vs. C09/C10 prescription as independent variable and sex and age as covariates in the subset of the population that had a hospitalization because of dyspepsia, gastritis/duodenitis or gastric/duodenal ulcer (ICD10: K25, K26, K29, K30).

Another possible bias may be due to patients receiving an acid inhibitor being more likely to receive other medications as well. This hypothesis was tested by computing the prevalence of C09/C10 prescriptions in people receiving and not receiving acid inhibitors, considering prescription of a non-steroidal anti-inflammatory drug (M01). Furthermore, we analyzed time to an anti-allergic prescription after a previous anti-acid one stratified by presence or absence of a M01 prescription.

Analyses were performed by Stata 13.1 (StataCorp, TX, USA) and R 3.1.3 (r-project.org). Figures were produced by Statistica 10.0 (StatSoft, OK, USA) and Excel 2010 (MicroSoft).

**Reporting summary.** Further information on research design is available in the Nature Research Reporting Summary linked to this article.

## Data availability

The data that support the findings of this study were made available to the authors of this study by all major Austrian compulsory health insurance companies under the ethics vote ECS 1134/2014. General Data Protection Regulations Restrictions apply to the availability of these data, which were used under license for this study. However, any researcher can access the data by obtaining an ethical approval from the regional ethical review board and thereafter addressing requests for data to DI Berthold Reichardt, representative for the involved Austrian compulsory health insurance companies.

## Code availability

Most analyses were conducted using standard procedures without special code development. For data extraction and calculation of prevalences, code can be obtained upon request to the corresponding author.

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

## Acknowledgements

We are most thankful to Dr. Sophia N. Karagiannis, King's College London, for English proofreading and editing our manuscript, to Dr. Rodolfo Bianchini, from the inter-university Messerli Research Institute, Vienna, Austria, for his scientific input, and to Mag. pharm. Maria Kundracikova, Hospital Pharmacy, General Hospital of Vienna, for her invaluable advice throughout this work. This study was kindly supported by the BGKK ("Burgenländische Gebietskrankenkasse") and a grant of the Austrian Science Fund FWF, SFB F4606-B28, to E.J.J.

## Author contributions

G.J. wrote the manuscript draft, contributed to data collection and performed the data analysis. M.K. performed the analysis, contributed analysis tools, contributed to conceiving and designing the analysis and to writing of the manuscript. E.U. and I.P.S. contributed to data analysis and writing of the manuscript. B.R. collected the data and contributed to the analysis design. G.J., I.P.S. and E.J.J. designed Fig. 1. E.J.J. conceived and designed the analysis, contributed to collection of the data, data analysis, writing of the paper, and critical revision of the manuscript.

## Additional information

**Competing interests:** The authors declare no competing interests. The funders played no role in the design of the study, in the collection, analysis or interpretation of data, or drafting or writing of the manuscript, in the decision to submit the paper for publication, and did not review or approve the manuscript before submission.

