## [Peer Review File · Nature Communications]

Reviewers' comments:

Reviewer #1, expert on food allergy (Remarks to the Author):

This manuscript reports the incidence rate of anti-allergic drug prescriptions (antihistamines and desensitisation therapy) after prescription of gastric acid inhibitors, using data from health insurance claim records in Austria. The authors report that patients prescribed gastric acid-inhibiting drugs are more likely to receive a subsequent prescription of anti-allergic drugs compared to patients without a prescription for gastric acid inhibiting drugs.

Major concerns:

1. The premise for examining the association between gastric acid-inhibiting drugs and anti-allergic drugs is that these drugs impair protein digestion and may therefore increase the likelihood of sensitisation to orally ingested allergens (foods and drugs). This question is of considerable interest and there are previous observational studies and mouse models that support that this might be the case (as described by the authors in their background section). However, the data set used in this manuscript does not appear to be appropriate to examine this hypothesis. The antihistamines and desensitisation therapy prescribed in this dataset are far more likely to have been used for treating allergic rhinitis rather than food or drug allergy, since (1) these conditions are far less common than allergic rhinitis, (2) patients with food or drug allergy do not routinely need to take antihistamines as the first line treatment is avoidance of the food/drug, while patients with allergic rhinitis will require antihistamines more often, and (3) desensitisation therapy for food allergy is still mostly a research tool and not in widespread use clinically in many areas of the world (it may be the case that it is already in clinical use in Austria, if so this needs to be stated in the manuscript). There is no specific consideration in the manuscript of the potential biological plausibility (or lack of plausibility) for an association between gastric acid inhibiting drugs and an increase in allergic rhinitis, and no discussion of these limitations of this outcome measure. A better outcome measure would be prescription of adrenaline autoinjectors, which are more specifically prescribed for severe food and drug allergies (also, less commonly, for insect sting allergy).
2. The statistical evaluation section of the manuscript is unclear and more information is required on how the incidence rates were calculated, particularly for those without an acid inhibitor prescription. The authors state that the denominator for calculating rates in those WITH an acid inhibitor prescription was the total person-years of follow up until either prescription of an anti-allergic drug or end of follow up (line 68). By contrast, the denominator for those WITHOUT an acid inhibitor prescription was the total person-years of follow up minus the person-years for those with an acid inhibitor prescription. This appears to imply that (1) the person-years of follow up after prescription of an anti-allergic drug in those with an acid inhibitor prescription may have been incorrectly included in the denominator here, and (2) that the denominator has been calculated differently for those without an acid inhibitor prescription, since the person-years of follow up for this group did not stop at the time of prescription of an anti-allergic drug. These two factors (if true) would have inflated the denominator and could contribute to the apparently lower incidence rate in this group.
3. An important limitation of this data is the lack of information on important potential confounders (other than age and gender) such as race/ethnicity and socioeconomic status. This needs to be carefully considered and discussed. Lack of data on confounding factors may be particularly important in the second analysis comparing those with antihypertensive and/or lipid modifying agents to those with acid inhibitor prescriptions. These two patient groups may well be very different in terms of factors such as ethnicity and socioeconomic status - both of which have also previously been shown to affect the risk of allergic disease.
4. There is a possibility that patients taking gastric acid inhibitors may develop an allergic response to the medication itself that leads to prescription for antihistamines (although I acknowledge that this may be very rare). It would be useful if the authors could comment in the discussion on whether this

may contribute to their findings.

Reviewer #2, expert on electronic health record studies (Remarks to the Author):

The authors use prescription data from health insurance claims in an Austrian population and examine rates of anti-allergy prescriptions among individuals who do and do not receive prescriptions for antacid medications. They find increased rates of allergy medication prescribing among individuals prescribed antacid medications. They attempt to mitigate potential bias related to prescribing patterns by using a subset of their population to ascertain whether individuals taking select cardiac-related medications have different baseline rates of anti-allergy medication use.

Leveraging large observational resources is an important approach to identifying unanticipated or novel adverse effects of medications. However, these observational studies have sources of confounding that can lead to biased observations. The study would be enhanced if the authors better controlled for potential sources of confounding using an approach such as propensity scores. Their findings would also be more compelling if they demonstrated that anti-allergy medication prescribing was linked to measures of potencies of antacid medications.

Major points:

Abstract:

1. The analyses do not provide evidence of “a causal relationship”, but rather demonstrate an association in an observational data set. I suggest replacing the word “causal”.

Methods:

1. (Page 5, line 73) Why was just a subset of the population used for the secondary analysis of individuals on statins and anti-hypertensives? Could the entire population have been used?
2. (Page 5, line 77) The authors should clarify that they are only looking at renin-angiotensin inhibitors, and not all anti-hypertensive medications, as these medications have specific side-effect profiles (e.g. cough) and contraindications (such as they are contra-indicated on women of reproductive age who may become pregnant).
3. (Page 5, line 83) Explain how “daily doses per year” is calculated. Does this mean that medications with more frequent dosing (i.e. a short half-life) receive a higher score? Or is it just the number of days an individual is on a medication?

Results:

1. The authors appropriately bring up the issue that users of acid suppression may be more likely to receive treatments overall. They should directly test this by determining the number of subsequent medication prescriptions among users and non-users of antacids.
2. An important assumption of these analyses is that users of antacids are similar to non-users with respect to the likelihood of subsequent exposure to allergy medications. Often this may not be true in observational studies. One commonly used approach to address this bias is to perform propensity score matching or adjustment, which adjusts for confounding factors predictive of the exposure. While it is not clear what variables may be available to the authors to construct a propensity score, variables could include other medications prescribed prior to the antacid drug, the indication for a drug and co-morbidities.
3. (Page 7, line 135): It is not clear that the results from the analyses stratified by “Longest applied acid inhibitor” and by “Daily acid inhibitor dose per year (DDD/y)” support acid inhibition as a mechanism of allergy. PPIs are the most potent acid-suppressing medications and would seemingly be expected to have the strongest association. In contrast, the primary mechanism of sucralfate is not acid suppression, yet this drug shows that strongest association with allergy medication. Similarly, while there is a clear trend with DDD/y, it is not clear whether longer duration simply reflects the use of weaker antacids. The DDD/y analyses should be stratified by type of acid inhibitor to better understand a dose response. Overall, both of these analyses would be more compelling if the authors

showed an association with a measure reflecting the acid-reducing potential of the drug and duration of exposure. If this measure were associated with allergy medications, it would be more suggestive that acid suppression may be the underlying mechanism.

Discussion:

1. The indication for why an antacid was prescribed is unknown. In some instances, antacids can be used to treat a cough due to reflux disease. A cough may also be due to allergy symptoms, and this could account for part of the association. The authors should note that the indications are not known and could contribute to the observed association.
2. An alternative explanation for the findings is that dyspepsia/gastric ulcers are associated with subsequent use of allergy medications, and that antacids are just a marker for this disease. One approach to test this would be to conduct the analyses only among individuals who carry a relevant gastro-intestinal diagnosis and then determine whether the association between antacids and allergy medications persists among this group. If this is not feasible, the authors should note this alternative explanation in the limitations.

Tables and Figures:

Table 3: Define "Daily acid inhibitor dose per year" in a footnote.

Minor points:

Methods:

1. Typo (page 4, line 62): I believe the text should say "...5-year age groups..."

PBP Replay to Reviewer #1, expert on food allergy (Remarks to the Author):*Major concerns:*

1. *The premise for examining the association between gastric acid-inhibiting drugs and anti-allergic drugs is that these drugs impair protein digestion and may therefore increase the likelihood of sensitisation to orally ingested allergens (foods and drugs). This question is of considerable interest and there are previous observational studies and mouse models that support that this might be the case (as described by the authors in their background section).*

We want to thank the reviewer for kindly stating our study question to be of “considerable interest”. We need to add that also human studies on selected cohorts were preceding our work. To emphasize the collected evidence, we now arranged a novel comprehensive illustration (Figure 1).

However, the data set used in this manuscript does not appear to be appropriate to examine this hypothesis. The antihistamines and desensitisation therapy prescribed in this dataset are far more likely to have been used for treating allergic rhinitis rather than food or drug allergy, since (1) these conditions are far less common than allergic rhinitis, (2) patients with food or drug allergy do not routinely need to take antihistamines as the first line treatment is avoidance of the food/drug, while patients with allergic rhinitis will require antihistamines more often, and (3) desensitisation therapy for food allergy is still mostly a research tool and not in widespread use clinically in many areas of the world (it may be the case that it is already in clinical use in Austria, if so this needs to be stated in the manuscript). There is no specific consideration in the manuscript of the potential biological plausibility (or lack of plausibility) for an association between gastric acid inhibiting drugs and an increase in allergic rhinitis, and no discussion of these limitations of this outcome measure. A better outcome measure would be prescription of adrenaline autoinjectors, which are more specifically prescribed for severe food and drug allergies (also, less commonly, for insect sting allergy).

The referee exactly extracted the novelty of our study better than we did, evidencing that PPIs and anti-ulcer drugs provoke a systemic Th2 bias not only important in eliciting food allergies, but in determining IgE-mediated allergy *per se*, which then altogether results in prescription of symptomatic allergy medications (in fact, also in diagnosed food allergy anti-histamines besides in severe cases adrenalin autoinjectors are prescribed as rescue medications in Austria). This more general insight is novel and plausible considering the collected evidence for the effects of PPIs and anti-ulcer drugs on innate and adaptive cells. We are therefore thankful for the referee’s comments which made us re-think the results and gave us the chance to improve the interpretation and truly understand the implications. The novel Figure 1 is a core element of this upgraded perception, as well as a novel paragraphs added in the introduction section and in the discussion, supported by novel references, see pages 3, 4 and 10 respectively.

2. The statistical evaluation section of the manuscript is unclear and more information is required on how the incidence rates were calculated, particularly for those without an acid inhibitor prescription. The authors state that the denominator for calculating rates in those WITH an acid inhibitor prescription was the total person-years of follow up until either prescription of

an anti-allergic drug or end of follow up (line 68). By contrast, the denominator for those WITHOUT an acid inhibitor prescription was the total person-years of follow up minus the person-years for those with an acid inhibitor prescription. This appears to imply that (1) the person-years of follow up after prescription of an anti-allergic drug in those with an acid inhibitor prescription may have been incorrectly included in the denominator here, and (2) that the denominator has been calculated differently for those without an acid inhibitor prescription, since the person-years of follow up for this group did not stop at the time of prescription of an anti-allergic drug. These two factors (if true) would have inflated the denominator and could contribute to the apparently lower incidence rate in this group.

We are sorry for the imprecision in the statement about calculation of person-years of follow up. We have explained the procedure in greater detail in the revised version, please see page 6. The denominator for those WITHOUT an acid inhibitor prescription was calculated as the total number of person-years of the insured population minus the person-years for those with an acid inhibitor prescription minus person-years under an anti-allergic prescription in those without prior acid inhibitor prescription. Note that otherwise the person-years for anti-allergic prescriptions in persons with a prior acid inhibitor prescription would have been subtracted twice.

3. *An important limitation of this data is the lack of information on important potential confounders (other than age and gender) such as race/ethnicity and socioeconomic status. This needs to be carefully considered and discussed. Lack of data on confounding factors may be particularly important in the second analysis comparing those with antihypertensive and/or lipid modifying agents to those with acid inhibitor prescriptions. These two patient groups may well be very different in terms of factors such as ethnicity and socioeconomic status - both of which have also previously been shown to affect the risk of allergic disease.*

It is correct that variables such as ethnicity and SES could in general confound relationships with allergy as the outcome. However, it has to be noted that Austria is quite homogenous concerning ethnicity with a virtually 100% Caucasian population. Variation in SES could be a factor due to differences in access to the health care sector. However, due to 97% coverage of the population by health insurance with no threshold for access to treatment for allergic diseases (costs are covered by the insurance) we think that there is little cause for concern that such confounders are relevant. We have, however, included this issue in the Discussion, see page 11. We were rather worried that contact with the health care sector as such, due to any other disease, could be a confounder, because case history taken by a physician may lead to detection and treatment of allergies that may remain unnoticed or at least untreated without such contact. Therefore, we included the control cohort treated for hypertension or dyslipidemia.

4. *There is a possibility that patients taking gastric acid inhibitors may develop an allergic response to the medication itself that leads to prescription for antihistamines (although I acknowledge that this may be very rare). It would be useful if the authors could comment in the discussion on whether this may contribute to their findings.*

Indeed, allergic sensitization to orally administered drugs resulting from anti-ulcer drug induced gastric pH modulation has been previously described by others and by our own group (Riemer AB et al, Clin. Exp. Allergy 2010). Moreover, other groups have demonstrated that sensitization may even more occur also to the PPI itself. PPIs are for instance also applied in the treatment of eosinophilic esophagitis, but an increasing number of studies proposes that they induce Th2 responses instead of

healing them, therefore replacing one evil by another. We thus agree with the reviewer that the possibility of acid inhibitor sensitization is a rare event only, and as suggested we commented on this in the Discussion, see page 9.

PBP Replay to Reviewer #2, expert on electronic health record studies (Remarks to the Author):

The authors use prescription data from health insurance claims in an Austrian population and examine rates of anti-allergy prescriptions among individuals who do and do not receive prescriptions for antacid medications. They find increased rates of allergy medication prescribing among individuals prescribed antacid medications. They attempt to mitigate potential bias related to prescribing patterns by using a subset of their population to ascertain whether individuals taking select cardiac-related medications have different baseline rates of anti-allergy medication use.

Leveraging large observational resources is an important approach to identifying unanticipated or novel adverse effects of medications.

We thank the reviewer for appreciating the importance of large observational studies. Based on our previous smaller observational studies in humans and our experiments in mouse models, the presented work aimed to study the connection between gastric pH modulation and allergic disease on an epidemiologically relevant scale.

However, these observational studies have sources of confounding that can lead to biased observations. The study would be enhanced if the authors better controlled for potential sources of confounding using an approach such as propensity scores. Their findings would also be more compelling if they demonstrated that anti-allergy medication prescribing was linked to measures of potencies of antacid medications.

We thank the reviewer for suggesting to demonstrate a link between potency of antacid medications and risk of subsequent anti-allergic medication. If restricting analysis to those receiving only one type of medication (PPI, H2RA, or Su – for PGE2 there were too few cases) there is a relationship between the risk for subsequent anti-allergic medication and potency, PPIs and H2RA have almost the same associated hazard ratio and their potency is indeed considered equal, and sucralfate has a somewhat lower potency in accordance with its lower hazard ratio: PPI HR=2.452; H2RA HR=2.401; Su HR=1.970. This information was added in the manuscript, see page 8.

Major points:

Abstract:

1. *The analyses do not provide evidence of "a causal relationship", but rather demonstrate an association in an observational data set. I suggest replacing the word "causal".*

This point is well taken. We have rephrased the wording as suggested, see pages 2 and 9.

Methods:

1. *(Page 5, line 73) Why was just a subset of the population used for the secondary analysis of individuals on statins and anti-hypertensives? Could the entire population have been used?*

Each county of Austria has its own general insurance company and some professions have their own insurance company, but our study has been approved for all Austrian insurance companies by the Steering Committee. When we had started the study over the whole Austrian population we became at a time concerned about the possibility that contact to the health care sector due to any disease could increase the chance for an allergic diagnosis and treatment, prompting the need for a second analysis including cohorts with control medications. Unfortunately, at this time point only one county (Burgenland) still had the data files for the total target period available.

2. (Page 5, line 77) The authors should clarify that they are only looking at renin-angiotensin inhibitors, and not all anti-hypertensive medications, as these medications have specific side-effect profiles (e.g. cough) and contraindications (such as they are contra-indicated on women of reproductive age who may become pregnant).

This information is indeed essential and we added this to the manuscript as suggested, see page 11.

3. (Page 5, line 83) Explain how "daily doses per year" is calculated. Does this mean that medications with more frequent dosing (i.e. a short half-life) receive a higher score? Or is it just the number of days an individual is on a medication?

Daily doses per year are number of days on a medication during follow up divided by years of follow up for each individual. If, for example, a medication is prescribed twice a day two doses make up one dose-day.

Results:

1. The authors appropriately bring up the issue that users of acid suppression may be more likely to receive treatments overall. They should directly test this by determining the number of subsequent medication prescriptions among users and non-users of antacids.

Although this is not what we intended, this is indeed an interesting suggestion for future studies. Strictly speaking this would involve inclusion of all medications. However, we determined the number of subsequent medication prescriptions among users and non-users of antacids for the prescription of antihypertensive drugs and statins. There is an indication that users of acid suppression have a higher proportion of such prescriptions (45%) as compared to non-users (27%). However, this relationship could be biased due to co-prescription of acid inhibitors for protection of NSAIDs related gastric symptoms. NSAIDs are often prescribed together with C09/C10 drugs for primary or secondary prevention of heart diseases.

2. An important assumption of these analyses is that users of antacids are similar to non-users with respect to the likelihood of subsequent exposure to allergy medications. Often this may not be true in observational studies. One commonly used approach to address this bias is to perform propensity score matching or adjustment, which adjusts for confounding factors predictive of the exposure. While it is not clear what variables may be available to the authors to construct a propensity score, variables could include other medications prescribed prior to the antacid drug, the indication for a drug and co-morbidities.

Unfortunately extraction of individual data other than age and gender was not supported by the ethics committee. Hence we have no data about co-morbidities or concomitant medications other than those mentioned. Therefore, propensity score matching or adjustment was not possible. We adjusted only for gender and age at first prescription. We mentioned this limitation in the revised version, page 11.

3. (Page 7, line 135): *It is not clear that the results from the analyses stratified by "Longest applied acid inhibitor" and by "Daily acid inhibitor dose per year (DDD/y)" support acid inhibition as a mechanism of allergy. PPIs are the most potent acid-suppressing medications and would seemingly be expected to have the strongest association. In contrast, the primary mechanism of sucralfate is not acid suppression, yet this drug shows that strongest association with allergy medication.*

This is an important comment that adds to the suggestion of reviewer #1 to address the relationship between potency of the different drug and risk for anti-allergic medication. The problem is that many receive more than one of these medications. If restricting analysis to those that received only one group of medications (PPIs, H2RA, Su – PGE2 was left out due to low numbers), sucralfate is the medication with the lowest associated hazard ratio. This indicates that Su was often prescribed as first drug but then, if not successful, was replaced by PPIs or H2RA. This then leads to an overestimation of the effect for Su.

However, sucralfate was still found to almost double the risk for subsequent anti-allergic prescription. This is likely due to the fact that sucralfate is an aluminum compound and acts as Th2 adjuvant by the oral and injected route. We have previously demonstrated this immune biasing effect in several studies (see most representative examples listed below).

Brunner R, et al. Aluminium per se and in the anti-acid drug sucralfate promotes sensitization via the oral route. *Allergy*. 2009 Jun;64(6):890-7. doi: 10.1111/j.1398-9995.2008.01933.x. Epub 2009 Feb 5.

Pali-Schöll I, et al. Anti-acids lead to immunological and morphological changes in the intestine of BALB/c mice similar to human food allergy. *Exp Toxicol Pathol*. 2008 Aug;60(4-5):337-45. doi: 10.1016/j.etp.2008.03.004. Epub 2008 Jun 4.

Brunner R et al. The impact of aluminium in acid-suppressing drugs on the immune response of BALB/c mice. *Clin Exp Allergy*. 2007 Oct;37(10):1566-73. Epub 2007 Sep 10.

To make the rationale more clear, we tried to highlight the Th2 adjuvant effect by sucralfate in the amended manuscript and added a most recent reference containing a detailed overview on the diverse mechanisms of antiulcer drugs to induce allergy (*The Effect of Digestion and Digestibility on Allergenicity of Food*. Pali-Schöll I, Untersmayr E, Klems M, Jensen-Jarolim E. *Nutrients*. 2018 Aug 21;10(9). pii: E1129. doi: 10.3390/nu10091129).

Moreover, in accordance with the perception of Ref 1 we must emphasize that the Th2 bias induced by these medications is not only confined to food allergens, but is a systemic effect, explaining the strength and specificity of the phenomenon of the observed subsequent anti-allergy prescriptions. We constructed a novel Figure 1 collecting all evidence for immune mechanisms how PPIs, H2 receptor blockers and sucralfate trigger Th2 immunity, by acting on innate and adaptive immune cells.

4. *Similarly, while there is a clear trend with DDD/y, it is not clear whether longer duration simply reflects the use of weaker antacids. The DDD/y analyses should be stratified by type of acid inhibitor to better understand a dose response. Overall, both of these analyses would be more compelling if the authors showed an association with a measure reflecting the acid-*

reducing potential of the drug and duration of exposure. If this measure were associated with allergy medications, it would be more suggestive that acid suppression may be the underlying mechanism.

We thank the reviewer for this suggestion and have added the new analyses, see pages 8 and 9.

Discussion:

1. The indication for why an antacid was prescribed is unknown. In some instances, antacids can be used to treat a cough due to reflux disease. A cough may also be due to allergy symptoms, and this could account for part of the association. The authors should note that the indications are not known and could contribute to the observed association.

We agree with the reviewer that in patients presenting with symptoms such as “cough” of unknown origin, some doctors may decide to try an empiric treatment using acid-inhibitors and antihistamines, but there is no statistics about this possibility. We have added this aspect among the limitations.

2. An alternative explanation for the findings is that dyspepsia/gastric ulcers are associated with subsequent use of allergy medications, and that antacids are just a marker for this disease. One approach to test this would be to conduct the analyses only among individuals who carry a relevant gastro-intestinal diagnosis and then determine whether the association between antacids and allergy medications persists among this group. If this is not feasible, the authors should note this alternative explanation in the limitations.

It is true that the diseases themselves for which antacids are prescribed might be associated with an increased risk for an allergic disease. There are unfortunately no diagnoses in the insurance claims data base. However, hospital admissions including ICD codes are available. We have done an analysis restricted to those individuals that had a hospital admission for dyspepsia or gastric/duodenal ulcer. The result was the same: HR 2.098 95% CI: 1.369-3.215, p=0.0007. We have included these results in the revised version on page 8.

Tables and Figures:

Table 3: Define "Daily acid inhibitor dose per year" in a footnote.

Done, please see Table 3 page 18.

Minor points:

Methods:

1. Typo (page 4, line 62): I believe the text should say "...5-year age groups..."

Corrected; please see page 5.

Reviewers' comments:

Reviewer #1 (Remarks to the Author):

The authors' response to comments has raised one additional question. In the response to Reviewer #2, Results Comment 1, the authors state that they compared the number of subsequent medication prescriptions for antihypertensive drugs and statins among users and non-users of antacids. They find that users of acid suppression are more likely to receive subsequent prescriptions for antihypertensive drugs/statins. This increases the concern that the other findings of the current study may in fact be driven by the fact that antacid users are subsequently more likely to receive treatments overall. I believe these results should be added to the manuscript, and discussed.

The authors state that this finding could be explained by co-prescription of acid inhibitors with NSAIDs together with C09/C10 drugs. Since date of prescription was collected as part of the study, presumably this possibility could be formally explored and either confirmed or eliminated based on date of prescription of each of the medications.

Reviewer #2 (Remarks to the Author):

The authors have addressed my major points.

Minor point:

Typo: (Page 6, line 85): I believe this sentence should read "...and the period between the beginning of follow-up..."

Reviewers' comments:**Reviewer #1 (Remarks to the Author):**

The authors' response to comments has raised one additional question.

In the response to Reviewer #2, Results Comment 1, the authors state that they compared the number of subsequent medication prescriptions for antihypertensive drugs and statins among users and non-users of antacids. They find that users of acid suppression are more likely to receive subsequent prescriptions for antihypertensive drugs/statins. This increases the concern that the other findings of the current study may in fact be driven by the fact that antacid users are subsequently more likely to receive treatments overall. I believe these results should be added to the manuscript, and discussed.

The authors state that this finding could be explained by co-prescription of acid inhibitors with NSAIDs together with C09/C10 drugs. Since date of prescription was collected as part of the study, presumably this possibility could be formally explored and either confirmed or eliminated based on date of prescription of each of the medications.

We thank the reviewer for this suggestion and in response to this concern we have been able to additionally obtain this data set from the health insurances covering all prescription to an NSAID (M01). Indeed, after considering co-prescriptions this difference vanishes. Overall, 39.2% of those with a prescription of an acid inhibitor followed by a C09/C10 prescription had also a NSAIDs prescription. Hence, those without NSAIDs but C09/C10 prescription after an acid inhibitor prescription comprise 27.2% compared to 26.8% C09/C10 prescription in those individuals without prior acid inhibitor prescription. As requested, we have added this information to the manuscript, please see page 11.

We sincerely hope that the manuscript is now to the reviewer's satisfaction.

Reviewer #2 (Remarks to the Author):

The authors have addressed my major points.

We appreciate the valuable input of the reviewer and are pleased to have improved our manuscript to their satisfaction.

Minor point:

Typo: (Page 6, line 85): I believe this sentence should read "...and the period between the beginning of follow-up...":

Corrected, please see page 6;

We thank the reviewer for the rapid response!

REVIEWERS' COMMENTS:

Reviewer #1 (Remarks to the Author):

The authors have adequately addressed my comments

Reviewers' comments:

Reviewer #1 (Remarks to the Author):

The authors have adequately addressed my comments.

We thank the reviewer for his valuable suggestions and are happy to have attended to their requests in a satisfactory manner.